# C3-Symmetric Ligands in Drug Design: When the Target Controls the Aesthetics of the Drug

**DOI:** 10.3390/molecules28020679

**Published:** 2023-01-10

**Authors:** Mirjana Antonijevic, Christophe Rochais, Patrick Dallemagne

**Affiliations:** Normandie Univ, UNICAEN, CERMN, 14000 Caen, France

**Keywords:** symmetry, C3, homo-trimeric, drug design

## Abstract

A number of proteins are able to adopt a homotrimeric spatial conformation. Among these structures, this feature appears as crucial for biologic targets, since it facilitates the design of C3-symmetric ligands that are especially suitable for displaying optimized ligand–target interactions and therapeutic benefits. Additionally, DNA as a therapeutic target, even if its conformation into a superhelix does not correspond to a C3-symmetry, can also take advantage of these C3-symmetric ligands for better interactions and therapeutic effects. For the moment, this opportunity appears to be under-exploited, but should become more frequent with the discovery of new homotrimeric targets such as the SARS-CoV2 spike protein. Besides their potential therapeutic interest, the synthetic access to these C3-symmetric ligands often leads to chemical challenges, although drug candidates with an aesthetic structure are generally obtained.

## 1. Introduction

Rotational symmetry of order 3, also called 3-fold rotational, radial symmetry, or simply C3, is widely present in nature and many organisms obey to this spatial organisation, where the same structural moiety is repeated 3 times around an axis. This spatial arrangement usually confers an aesthetic aspect to these organisms. It is probably the reason why this kind of symmetry is, frequently and for a long time, mimicked by humans in a large number of artistic or functional realisations (Figure 1).

Concerning chemistry, C3-symmetric molecules have been widely used for a number of years, as well as compounds with two-fold rotational symmetry, especially in asymmetric catalysis and chiral recognition [1,2]. As C3-organisms and objects, C3-molecules also exhibit a structural moiety that is repeated three times around a rotational axis and can adopt either an acyclic, exocyclic, macrocyclic or bicyclic topology (Figure 2). Sometimes, this C3-symmetry can further be complicated by the presence of a second perpendicular rotational axis, yielding D3 symmetric compounds [3].

Within the field of medicinal chemistry, however, very few marketed drugs display a C3-symmetry, especially when compared with C2-drugs, for which the structure generally corresponds to the homodimeric status of their therapeutic target (e.g., symmetric diol inhibitors of HIV pol-protease, ombitasvir, daclatasvir and pibrentasvir against HCV NS5A protein, chlormethine, chlorambucil as DNA alkylating nitrogen mustards, mitoxantrone against topoisomerase II…) [4]. Nonetheless, these few C3-drugs that are currently present on the pharmaceutical market, were not specifically designed to be symmetric and consequently do not exploit this structural particularity in their mechanism of action. The interest in C3-drugs is widely growing, especially due to the validation of homotrimeric proteins as therapeutic targets (e.g., TNFα receptors, P2X receptors, SARS-CoV2 spike protein). Today, some C3-small molecules and biologics are designed with the use of old drugs that function as tripodal scaffolds in the synthesis of novel agents. 

This review will focus on the currently marketed C3-drugs and will then provide an update on novel C3-symmetric agents, specifically designed in order to bind appropriate molecular targets of therapeutic interest.

## 2. Currently Marketed C3-Drugs

To the best of our knowledge, only seven C3-drugs are available on the market today (Figure 3) [5].

Altretamine and thiotepa are considered as poly-alkylating agents of DNA and used in the treatment of breast, ovarian and bladder cancer. Paraldehyde is believed to decrease the release of acetylcholine in response to neuronal depolarization and to block neuromuscular transmission. It is used as a hypnotic, anticonvulsant agent and against cough. Ritiometan is an antimicrobial agent used in nasal sprays against rhinitis. Tromethamine is an alkaline compound and consequently used for the prevention and correction of metabolic acidosis. Methenamine and amantadine belong to the adamantane family, as well as rimantadine, memantine, saxagliptine and vidagliptine, but are the only C3-symmetric representatives of this series. Methenamine is an antimicrobial agent that releases ammonia and formaldehyde, and was recently proposed as an alternative to prophylactic antibiotics for the treatment of recurrent urinary tract infections in women [6]. Finally, amantadine is an old drug used as an alternative antiparkinsonian agent and against flu. In the latter indication, amantadine behaves as an influenza A virus matrix 2 (M2) proton channel inhibitor. As for the other C3-drugs, its symmetry is not requested for its activity and could instead appear as detrimental to its binding to the tetrameric M2 protein. It has, however, been demonstrated that, due to its rigidity, amantadine loses very few conformational entropies when it binds to M2, which, together with its very smooth surface, allows it to rotate quickly around its C-3 axis and to properly inhibit protons entering the virion and enabling the latter to become mature [7].

## 3. Recent Works concerning Novel C3-Symmetric Drugs

Today, C-3 symmetric compounds are more frequently deliberately designed to ensure they possess specific properties suited to therapeutic targets that are themselves homotrimeric proteins. These ligands are built starting from C3-tripodal cores. The ones that we have identified as being more specific are: (1) adamantanes, (2) benzene rings, (3) triazines, (4) aurintricarboxylic acid, (5) triPEGnitromethane derivatives, (6) nitrilotriacetic acid derivatives, or (7) cyclic peptides (Table 1).

### 3.1. Adamantane-Based Dendrons for Improving Ligand–Target Interactions

Dendrons are polymeric macromolecules composed of multiple branched monomers radially emanating from a central core [8]. These dendritic scaffolds are often used as carrier systems to deliver drugs or imaging labels chemically grafted to the dendrimer surface. Since tripodal C3-symmetric systems play an important role in molecular recognition, some adamantane-based dendrons have been synthesized and coupled to drugs to improve ligand–target interactions. P140 is a 21-mer peptide capable of decreasing autoimmune manifestations of systemic lupus by binding to a chaperone protein, and downregulating HSPA8 and autophagic flux. This leads finally to a decrease in the production of circulating antibodies. The activity of the tripodal coupled peptide was preserved and, therefore, multivalency was achieved thanks to the adamantane core, which appeared to be beneficial to the therapeutic activity [9].

### 3.2. Benzene Rings

Several C3-symmetric compounds use a tri-substituted benzene ring as a scaffold.

#### 3.2.1. C3-Opioids as DNA Condensation Agents

DNA condensation agents are of considerable interest; their potential use as non-viral vectors to deliver genes is today emphasized by the development of gene therapy. The non-covalent recognition scaffolds designed to exert this activity, generally, are cationic derivatives that neutralize the anionic charges of DNA phosphate groups. Thus, some DNA surface-binding agents, especially those exhibiting a charge of 3^+^ or more, have been shown to be capable of aggregating nucleic acids, including polyamine derivatives such as triplatin (Figure 4) [10,11,12]. As a continuation of this work, novel DNA condensation agents have been obtained, replacing the inert tripodal cobalt scaffold by mesetylene and amines by morphine, heterocodeine or oripavine. The resulting C3-symmetric opioids highly bound and condensed DNA, while their C1- and C2-congeners did not, accounting for a correspondence between the two kinds of symmetry affecting DNA on the one hand and these ligands on the other.

#### 3.2.2. Tris Triazole Compounds as G-Quadruplex Stabilising Ligands

Telomerase is frequently over-expressed in some cancers, leading to the synthesis of telomeric DNA repeats which are involved in cellular immortality [13,14]. The inhibition of telomerase can be achieved through the sequestration of the 3′ end of telomeric DNA into the G-quadruplex. Stabilisation of the latter by appropriate small molecules is consequently a promising approach against some cancers. The structural requirements needed to achieve this goal appear to be a planar aromatic scaffold and cationic side chains. Among the agents that adhere to this rule, C3-symmetric tris-triazole derivatives were described and obtained through click-chemistry, and were found able to stabilise the G-quadruplex in a selective manner towards duplex DNA (Figure 5). The three symmetric substituents of the ligands seemed capable of interacting; each one appeared in an individual groove of the quadruplex target [15].

#### 3.2.3. Tripodal Nitrogen Mustards for Aggregation Induced Emission and DNA Alkylation

Aggregate-induced emission (AIE) relies on the fluorescent properties displayed by hydrogels upon aggregation of weakly emissive single molecules. To achieve this aggregation, several tripodal gelators have been described. They facilitate the detection and separation of various ions (Fe^3+^, CN^−^, H_2_PO_4_^−^) [16,17] and aldehydes [18]. As a continuation of this work, it was recently described that tripodal nitrogen mustard (Figure 6) can be used for imaging cancers cells by fluorescence and, in the same way, it can exert cytotoxic activity through DNA bis alkylation. The C3-symmetry of the ligands seems again important for DNA recognition and binding. Such compounds are of particular interest within theranostic approaches.

#### 3.2.4. Benzenetricarboxamide as a Neurotrophic Agent against Neurodegenerative Diseases

LM22A-4 is a known activator of the neurotorophin tyrosine kinase receptor B (TrkB) [19]. The TrkB receptor seems to play a leading role not only in neuronal survival, axonal and dendritic growth, and plasticity, but also in adult neurogenesis [20,21,22,23]. Moreover, in many neurodegenerative disorders (ND) [24,25,26,27] it is proven that the function of the TrkB receptor is impaired. LM22A-4 has been developed as an analogue of the physiological TrkB ligand, the brain-derived neurotrophic factor (BDNF); more specifically, for its loop II [19]. The main idea in its development was the generation of a pharmacophore hypothesis based on the sub region b on the BDNF loop II, [19] which includes Ser94, Lys95 and Lys96 residues (Figure 7A). Its C3 symmetry was achieved in order to mimic these three residues (Figure 7B). However, if we review the other molecules from the same family of BDNF mimetics (LM22A dataset [19]) we can see that those molecules do not possess C3 symmetry. These findings could indicate that this symmetry was achieved by chance. In terms of its activity, LM22A-4 was shown to be active in several cell models of Alzheimer’s, Huntington’s [28], and Parkinson’s diseases at a concentration of 0.5 μM, [29] it also reduced the development of tissue injury associated with spinal cord trauma, [30] promoted survival of cultured retinal ganglion cells (RGCs), led to increased survival of RGCs in vivo after an optic nerve ischemia, [31] and ameliorated biochemical and functional abnormalities in a mouse model of Rett syndrome [32]. Even though LM22A-4 was developed as the BDNF mimetic, some findings are actually showing that this molecule is an indirect activator of the TrkB receptor rather than the direct one [33,34]. Despite these facts, we cannot discard the protective effect that this molecule possesses in ND.

#### 3.2.5. Benzoxazine Ureas as Synthetic Chloride Transmembrane Transporters

Artificial transport systems may be useful in a wide range of «channelopathies» such as epilepsy, cystic fibrosis, myotonies, etc. [36]. Among them, tripodal synthetic ionophores using a C3-symmetric benzoxazine core, bearing three ureido moieties, were shown as able to recognize anions and to exchange them with NO_3_^−^ ions through liposomal membranes (Figure 8). These benzoxazine ureido compounds appear, in size and composition, to be able to correctly interact with the liposomal membrane.

#### 3.2.6. Trivalent Hemagglutinin Inhibitors against Flu

Hemagglutinin is a viral adhesion protein responsible for the attachment of influenza A virus to sialic acid molecules of the host cell membrane. Inhibiting this protein–protein interaction could be synergistic with the inhibition of the viral neuraminidase exerted by zanamivir or oseltamivir which block the viral spread in the organism this enzyme is responsible for [37]. Hemagglutinin is a homotrimeric receptor and this multivalency has high affinity with which the virus binds to the host cell surface glycans. To overcome this key step in the flu infection, trivalent hemagglutinin inhibitors have been conceived by mimicking the C3-symmetry of the protein (Figure 9) [38]. 

The tripodal core of these inhibitors mainly contains a trisubstituted benzene ring, displaying glycan moieties in the three binding sites of the trimeric hemagglutinin. Alternatively, oleanic acid, a pentacyclic triterpene, can replace the glycans to inhibit the protein–protein interaction between hemagglutinin and sialic acid [39]. Glycan and oleanic acid moieties are often introduced on the benzene tripodal core using click chemistry and triazole formation. 

### 3.3. Triazine Derivatives as Tripodal Scaffolds

1,3,5-triazine is, as well as benzene, a C3-symmetric core but also a π-deficient heterocycle. It is, therefore, liable to engage in face-to-face interactions, contrarily to T-shape edge-to-face interactions of benzene [40]. 

#### 3.3.1. Trisusbtituted Triazines as Antiviral Triplet Drugs

Carbohydrates at the cell surface are pivotal to the recognition of host cells by viruses. Among them, glycocalyx, made of glycolipids, glycoproteins and proteoglycans are homo oligomeric units which often organize themselves as C3-symmetrical receptors. To interact with the latter, 1,3,5 homosubstituted triazines have been prepared, and some of them displayed potent anti-herpes simplex virus type 1 properties [41,42]. The lack of symmetry in these compounds led to the loss of activity.

#### 3.3.2. Star-Shaped Tripodal Triazine-Related β-Lactams with Antioxidant and Antibacterial Properties

Within the frame of a hybridization approach, lying in the synthesis of structural compromises between C-3 symmetrical triazine derivatives with antineoplastic properties and β-lactams with antibacterial properties, some tripodal triazines have been synthesized and evaluated. Their activities were enhanced with good inhibitory behaviour against human leukaemia cell line and antioxidant properties as radical scavengers [43]. It is not clear how these β-lactams take advantage of their C3-symmetry in their biological activities.

### 3.4. Aurintricarboxylic Acid as a Potent Allosteric Antagonist of P2X Receptors

P2X receptors are ATP-gated cation channels, among which P2X3R, particularly, is involved in sensory neurotransmission and its antagonists could be used against neuropathic pain. A high throughput screening of a large chemical library, aiming at selecting compounds for their capacity to antagonize a P2X3R mediated response, recently identified aurintricarboxylic acid as a nanomolar-potent allosteric antagonist of these receptors. Aurintricarboxylic acid is a trisalicylmethane acid derivative and its C-symmetry appears to be closely implied in these properties since P2X3 receptors are frequently assembled as an homotrimeric architecture of subunits [44].

### 3.5. triPEGnitromethane Derivatives for AIDS Treatment

HIV-1 possesses trans-membrane glycoproteins which play a significant role in the membrane fusion process of HIV-1 with the host cells and the entry of the pathogen into the lymphocyte cytoplasm. Among these glycoproteins, the *N*-terminal helix N36 and the C-terminal one C34 of gp 41 organize themselves as trimeric structures, N36 being surrounded by C34 in an antiparallel hairpin fashion. 

An innovant anti-HIV strategy is involved in the synthesis of tripodal peptides mimicking the natural trimers of C34 and N36 to constitute either fusion inhibitors or a synthetic antigen to induce the production of antibodies directed towards HIV-1 fusion proteins [45]. To mimic the trimeric architecture of C34, trihydroxypropylnitromethane was used as a tripodal core. Each of its three arms were engaged in a hydrophilic polyethylenglycol chain of equal length with a cysteine ligation site at its ends for coupling with three thioester derivatives of C34, in which a triplet repeat of arginine and glutamic acid was further added to increase aqueous solubility [46,47].

The synthesized C34 trimer analogue strongly binds to the N36 trimer, inhibits fusion of the HIV-1 membrane and those of the host cell in a 100-fold greater manner than the corresponding C34 monomer, and could be effective against enfuvirtide-resistant strains.

Furthermore, the C34 trimer was used to induce neutralizing antibodies directed towards gp41 and could constitute a novel class of HIV-1 vaccine.

### 3.6. Nitrilotriacetic Acid Derivatives

Nitrilotriacetic acid is an aminopolycarboxylic acid. It is a chelating agent, which forms coordination compounds with metal ions. It has been widely used in industry.

#### 3.6.1. Nitriloacetic Acid as a Scaffold for a Synthetic Antigen Inducing Neutralizing Antibodies against HIV

Nitriloacetic acid was used, similarly to triPEGhydroxypropylnitromethane, as a tripodal scaffold for the synthesis of HIV-gp41-derived peptides that mimic N36 and can be used as fusion inhibitors and in vaccines. To achieve this synthesis, the carboxylic acid groups of nitriloacetic acid are involved in three hydrophilic linker chains, with each bearing a terminal glycolaldehyde ester, which constitutes a ligation site. Each of them is prone to selectively reacting with a peptidic chain analogous to N36 which is conjugated with the tripodal core using a thiazolidine ligation strategy. The synthetic N36 trimer obtained forms a highly structured trimeric α-helix which finally perfectly mimics the viral glycoprotein N36. The latter is used either to inhibit the HIV-1 fusion step with host cells or to immunize mice and to produce antibodies directed against the viral fusion membrane and able to neutralize HIV-1 entry in cells [48,49].

#### 3.6.2. Nitrilotriacetic Acid Derivatives as Copper Chelators within Wilson’s Disease Treatment

Wilson’s disease is an orphan disease in which copper metabolism is altered, leading to its accumulation in hepatic cells, where it produces toxic hydroxyl radicals. D-Penicillamine is today the most widely used treatment of Wilson’s disease [50]. In some recent studies, the focus of the design of novel drugs against this disease are the proteins involved in the homeostasis of Cu(I), such as metallothioneins. The latter engage a complex with Cu(I) through three cysteine thiolates which coordinate the metallic cation. In order to mimic this interaction with Cu(I), some pseudopeptidic nitriloacetic acid derivatives, whose carboxylic acid groups were engaged in peptidic bounds with three cysteine residues or D-penicillamine moieties, have been synthesized [51]. The C3-symmetric core of these pseudopeptides acts as a tripodal platform, whose three thiolate groups are perfectly orientated to selectively coordinate copper. These chelators constitute promising agents for intracellular copper detoxification.

### 3.7. C3-Symmetric Cyclic Peptides as CD40L Mimetics

CD40 is a tumour necrosis factor receptor (TNF-R) whose interaction with its ligand CD40L, a glycoprotein, is essential for humoral and cellular immune responses [52].

Selective blockade or activation of this pathway can lead to treatments against immunologically diseases or cancers. CD40L monomers self-assemble according to a threefold symmetry axis and yield homotrimers that can each bind three receptors CD40. Mimicking CD40L homotrimers with small synthetic peptides, exhibiting a C3 symmetry, leads to perturbations in protein–protein interactions between CD40 and CD40L. This property seems able to be extended to other TNF-R family members [53].

## 4. Conclusions

Homotrimeric structures of proteins of potential therapeutic interest seem to be insufficiently exploited considering the relatively poor number of tripodal drugs. Yet, the number of proteins able to organize themselves under such a functional tertiary conformation is high, with emblematic examples (TNFα, HIV gp141, Flu virus hemagglutinin, etc.). The spike protein of SARS-CoV-2 constitutes another example with concrete application in the design of stable trimer vaccines with improved immunogenicity towards this cell entry glycoprotein [54,55]. However, small agents could also benefit from this strategy and especially protein–protein interaction perturbators. In this respect, artificial intelligence, especially machine learning methods for homo-trimeric protein interface residue pairs prediction, could be of critical assistance [56].

## Figures and Tables

**Figure 1 molecules-28-00679-f001:**
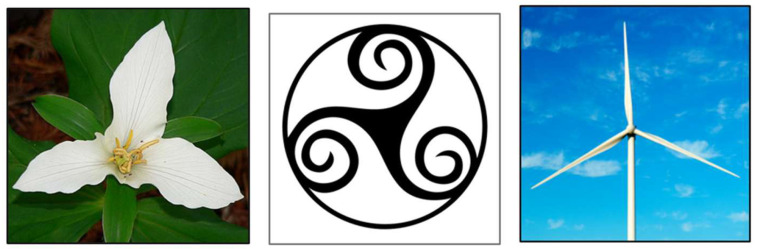
Left to right: *Trillium ovatum* flower; celtic symbol triskell; windmill.

**Figure 2 molecules-28-00679-f002:**
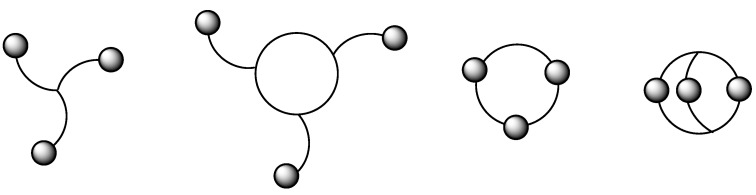
Representation of the different C3 topologies [3].

**Figure 3 molecules-28-00679-f003:**
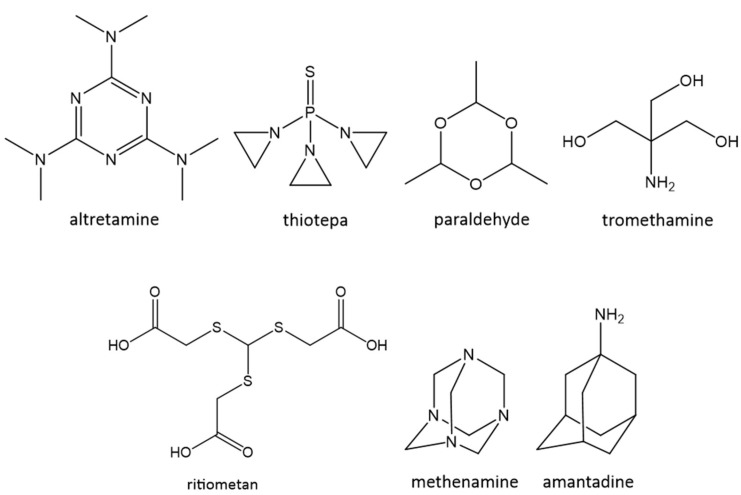
Structural representation of the currently marketed C3-drugs.

**Figure 4 molecules-28-00679-f004:**
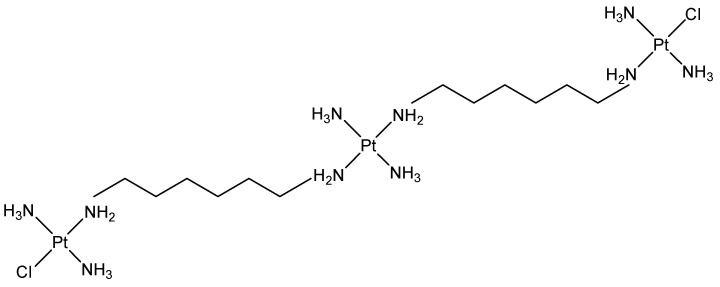
Structural representation of the triplatin.

**Figure 5 molecules-28-00679-f005:**
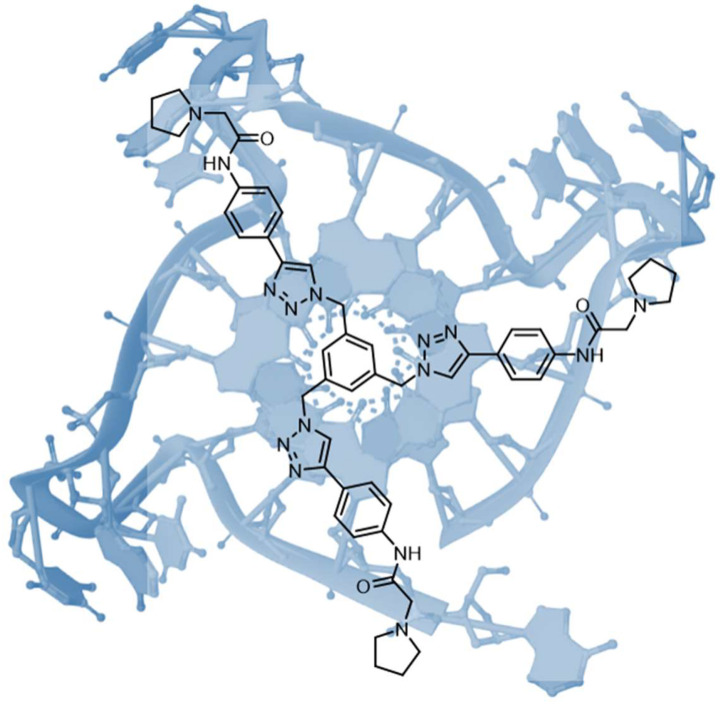
Representation of the C3-symmetric tris-triazole derivate bound to the parallel human telomeric quadruplex structure.

**Figure 6 molecules-28-00679-f006:**
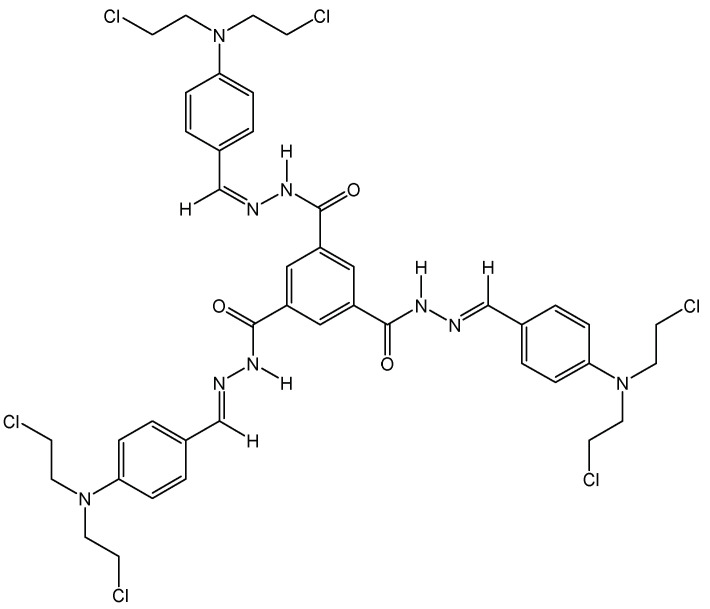
Structural representation of the tripodal nitrogen mustard.

**Figure 7 molecules-28-00679-f007:**
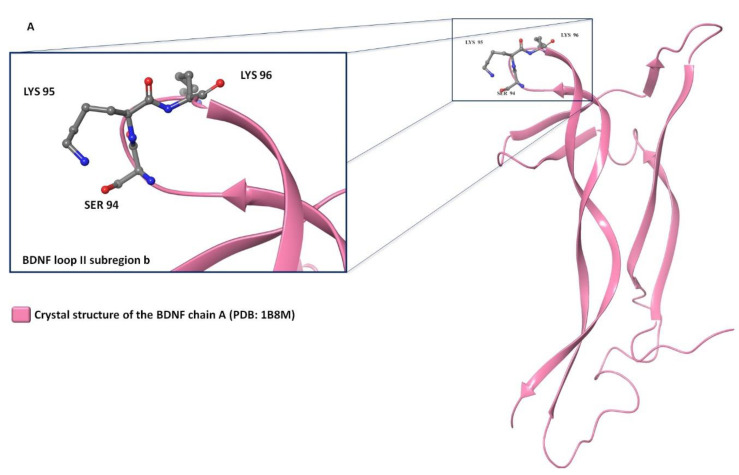
(**A**) Representation of the BDNF chain A crystal structure (PDB: 1B8M [35]) and its loop II. (**B**) Representation of the LM22A-4 and its alignment with the sub region b on the BDNF loop II.

**Figure 8 molecules-28-00679-f008:**
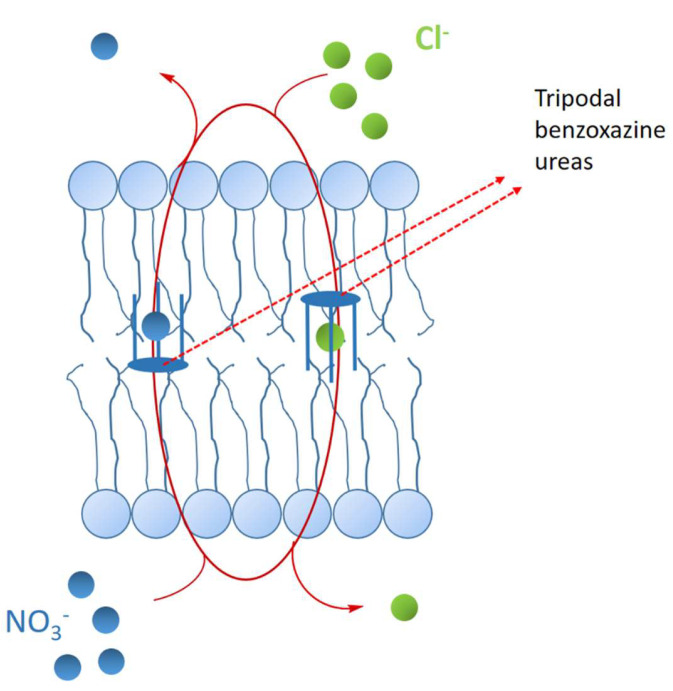
Representation of the C3-symmetric benzoxazine ionophores exchanging the anions trough liposomal membrane.

**Figure 9 molecules-28-00679-f009:**
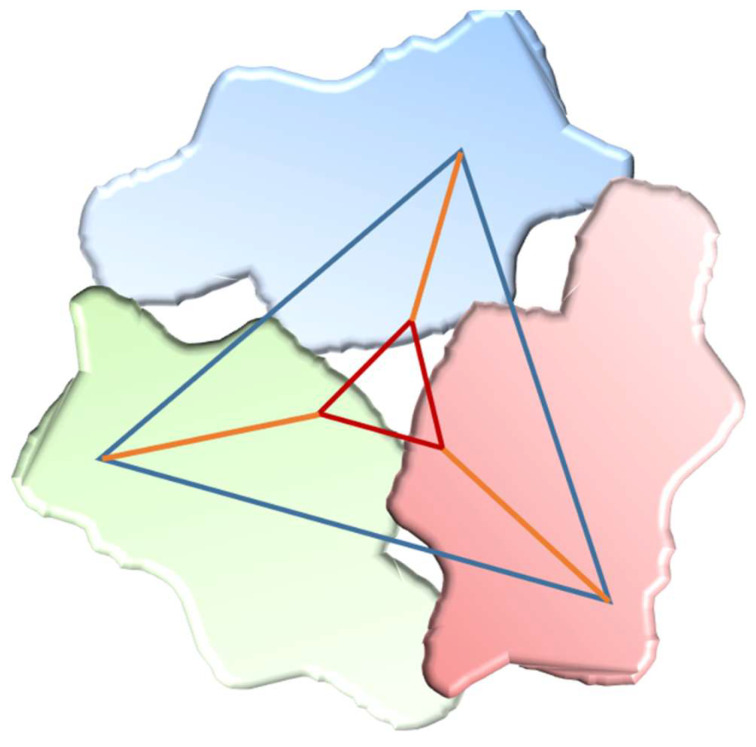
Representation of the hemagglutinin H5 trimer with distances highlighted between the binding sites.

**Table 1 molecules-28-00679-t001:** Examples of C3-symmetric ligands and their target.

C3-Tripodal Core	Target	Ligand Structure and Therapeutic Use	Examples
3-1-Adamantanes	HSPA8/ HSC70	Lupus	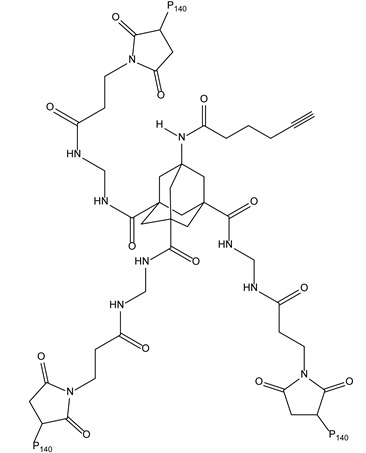
3-2-Benzene rings	DNA	3-2-1-C3-opioids as DNA condensation agents	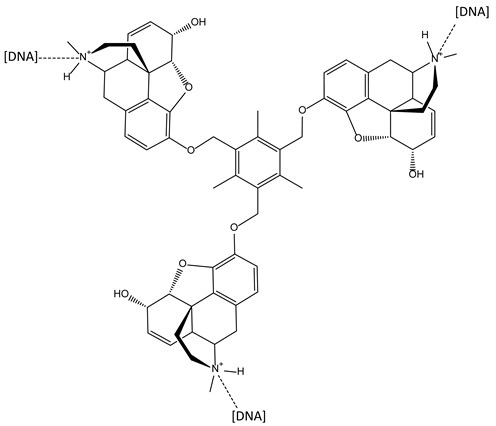
	DNA	3-2-2-Tris triazole compounds as G quadruplex stabilizing ligands	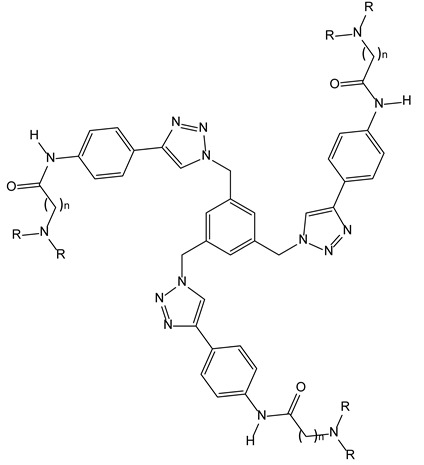
3-2-Benzene rings (continued)	DNA	3-2-3-Tripodal nitrogen mustards for aggregation induced emission and DNA alkylation	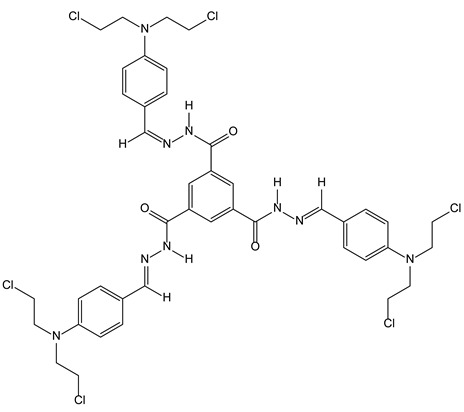
	TRKB R	3-2-4-Benzene tricarboxamide as neurotrophic agent against neurodegenerative diseases	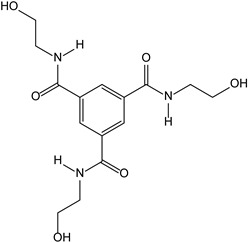
3-2-Benzene rings (continued)	Neuronal mem-branes	3-2-5-Benzoxazine ureas as synthetic chloride transmembrane transporters	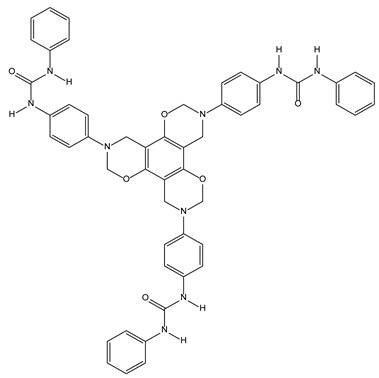
	Influenza virus HA	3-2-6-Trivalent hemagglutinin inhibitors against flu	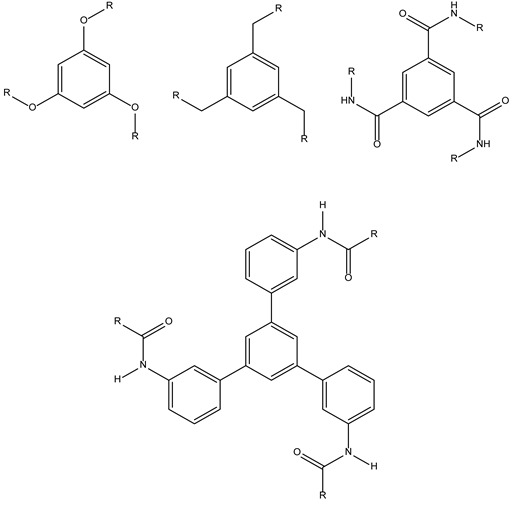
3-3-Triazines	Virus Carbohydrates	3-3-1-Anti-HSV-1	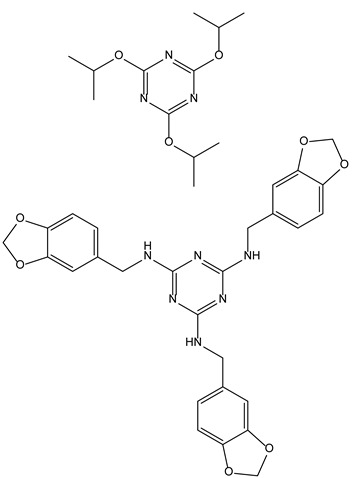
	Unkno-wn (pheno-typic test)	3-3-2-leukemia	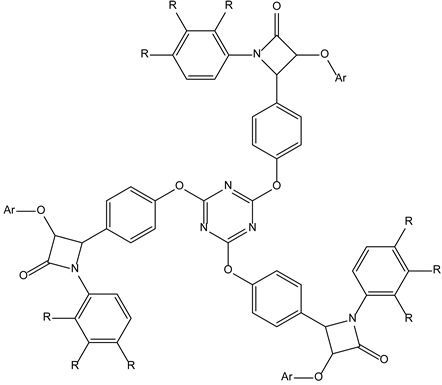
3-4-Aurintricarboxylic acid	P2X R	Chronic pain relief	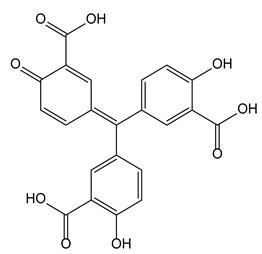
3-5-triPEGnitrométhane derivatives	Synthetic protein	AntiHIV antibody induction for fusion inhibition	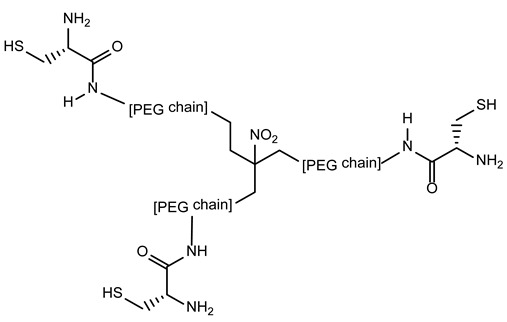
3-6-Nitrilotriacetic acid derivatives	Synthetic protein Cu(I)	3-6-1-AntiHIV antibody induction for fusion inhibition	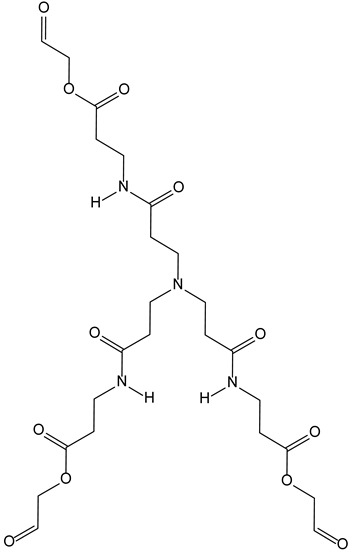
3-6-Nitrilotriacetic acid derivatives		3-6-2-Wilson’s disease	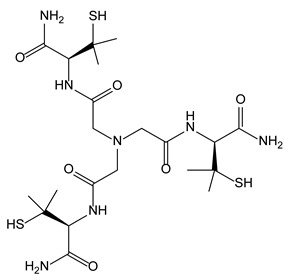
3-7-Cyclic peptides	TNFα R	CD40 ligand mimetics with anticancer activity	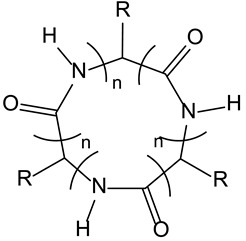

## Data Availability

Not applicable.

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
