# Peer review of "C3-Symmetric Ligands in Drug Design: When the Target Controls the Aesthetics of the Drug"

_molecules, 2023, doi:10.3390/molecules28020679_

Round 1

Reviewer 1 Report

Very interesting review article.

I only have 2 suggestions:

in the introduction I would just mention the other kind of symmetry that can be found in medicinal chemistry and add some references for review about compounds following these kinds of symmetry.

maybe it is linked to the submission process, but I would prefer to see structures along the review articles and not in a table. I could see that the table was following the structure of the next paragraphs, but it would be easier to follow if the structures were close to the corresponding subsection.

Author Response

We would like to sincerely thank the reviewer for his consciencious work.

Concerning some examples of C2-symmetric drugs, we have added the following sentence in the introduction part with an appropriate reference (GL Patrick In An introduction to Medicinal Chemistry, 6th edition, Oxford University Press: "e.g. symmetric diol inhibitors of HIV pol-protease, ombitasvir, daclatasvir and pibrentasvir against HCV NS5A protein, chlormethine, chlorambucil as DNA alkylating  nitrogen mustards, mitoxantrone against topoisomerase II...[4]"

We remain to the disposal of the editor concerning the eventual replacement of  table 1 by appropriate figures.

Reviewer 2 Report

This review conveys a wide set of 3-fold rotational symmetric molecules that serve or may serve as drugs by binding to DNA, proteins or even glycans. The paper is well written, it is novel and meaningful. Indeed, it does have several kinds of examples, both regarding the types of molecules and the targets. Thus, the paper may be published. Just few minor points should be checked: 

- The tables containing the molecules' structures are too big. Perhaps there is no other way of showing them, maybe the authors can try to think otherwise and do it differently. 

-  Resolution of Figures 7 and 8 is not the best, the authors should consider to improve it. 

- In line 160, nitrate group should be well written, the negative charge and the stoichiometry of oxygen is not well represented. 

- Both lactams (lines 193 and 198) and helix (line 252) seem to lack a greek letter, namely, beta and alpha respectively. It does not appear in the PDF file. 

Author Response

We would like to sincerely thank the reviewer for his consciencious work.

We remain to the disposal of the editor concerning the eventual replacement of  table 1 by appropriate figures.

New figures with better resolution will be provided.

The nitro group, as well as the greek letters appear well written before the pdf conversion. The problem will be fixed with the editor.

Reviewer 3 Report

The MS by Antonijevic et al. deals with C3-symmetric compounds, as ligands for a growing number of protein and DNA targets of pharmacological interest. Starting from marketed C3-drugs, they then discuss available data concerning various families of compounds with this symmetry, highlighting their superiority over related molecules for selected targets. The topic reviewed is timely and of interest in drug discovery and biochemistry. The MS is well written and structured. The table and figures are pertinent.

I only have some minor suggestions for the authors consideration:

-The abstract and introduction are centered in protein targets, for which the improved activity of C3-symmetric ligands is straightforward, considering the occurrence of trimeric oligomers. However, some of the compounds included in the MS actually interact with DNA (C3-opioids, Tris triazole compounds or Tripodal nitrogen mustards). This perhaps should be mentioned somewhere in the abstract and considered in the introduction, stating why and/or in which cases C3-symmetric compounds present higher activity as DNA binding ligands than other types of molecules.

-Section 2: The title “discussion” does not seem appropriate. I suggest replacing it by that in 2.1 (“Currently marketed C3-drugs”), and merging these sections accordingly. In addition, the implications of the C3-symmetry for recognition are only discussed in the case of amantadine. I suggest doing the same for the other molecules or stating that there is no information available in the literature. If the latter, perhaps the authors could formulate some hypotheses based on the nature of the targets, and consider shifting adamantine to the beginning of the paragraph, as its connections with the theme of the review would be stronger than those of the other compounds.

-Section 3: The title may be reconsidered, to better reflect the content.

-Sections 3.1.3 and 3.1.5, 3.2.2 are difficult to read and the advantage of the C3-symmetry is not obvious in those cases.

-Figure 5 is not mentioned in the main text.

-Line 217: a figure 13 is mentioned but I cannot find it in the MS.

-I suggest adding references for some of the general statements in: the last paragraph of page 2, last paragraph of page 10, last paragraph of page 14, fist 2 paragraphs in section 3.5 and in sections 3.6.1, 3.6.2 and 3.7.

-Minor English style mistakes should be fixed.

Author Response

We would like to sincerely thank the reviewer for his consciencious work.

Concerning the abstract, we have added the following sentence: "Additionally, DNA as a therapeutic target, even if its conformation into a superhelix does not respond to a C3-symmetry, can also take advantage of these C3-symmetric ligands for better interactions and therapeutic effect. "

The title "discussion" was removed.

The adamantine paragraph was moved at the beginning of the manuscript and some elements concerning the importance of the C3-symmetry for recognition of the targets by the ligands have been highlighted (please, see underlined sentences).

The title for section 2 was replaced by "Recent works concerning novel C3-symmetric drugs"  and the words "more and more frequently" were added before deliberately.

Figure 5 has been mentioned and figure 13 was removed.

Some new references have been added as requested (please see highlighted references).

The english language has been polished.